# MicroRNA-379 Modulates Prostate-Specific Antigen Expression Through Targeting the Androgen Receptor in Prostate Cancer

**DOI:** 10.3390/cancers17193245

**Published:** 2025-10-07

**Authors:** James R. Cassidy, Margareta Persson, Gjendine Voss, Kira Rosenkilde Underbjerg, Tina Catela Ivkovic, Anders Bjartell, Anders Edsjö, Hans Lilja, Yvonne Ceder

**Affiliations:** 1Division of Translational Cancer Research, Department of Laboratory Medicine, Lund University, 22381 Lund, Sweden; james.cassidy@med.lu.se (J.R.C.); gjendine.voss@med.lu.se (G.V.); tina.catela_ivkovic@med.lu.se (T.C.I.); 2Division of Oncology, Department of Clinical Sciences, Lund University, 22381 Lund, Sweden; margareta.persson@med.lu.se; 3Department of Urology, Skåne University Hospital, 20502 Malmö, Sweden; anders.bjartell@med.lu.se; 4Department of Clinical Genetics and Pathology, Laboratory Medicine, Medical Services, Region Skåne, 22185 Lund, Sweden; anders.edsjo@med.lu.se; 5Departments of Pathology and Laboratory Medicine, Surgery and Medicine, Memorial Sloan Kettering Cancer Center, New York, NY 10065, USA; hans.lilja@med.lu.se; 6Department of Translational Medicine, Lund University, 20502 Malmö, Sweden

**Keywords:** microRNA, oncogenic and tumor-suppressive miRNAs, prostate cancer, metastases

## Abstract

**Simple Summary:**

We have previously found that microRNA-379 (miR-379) may reduce the metastatic spread of prostate cancer to bone. In this study, we discovered that miR-379 affects how much prostate-specific antigen (PSA), the clinically used prostate cancer marker, is released by prostate cancer cells. However, miR-379 does not control PSA directly. Instead, our results indicate miR-379 reduces the levels of the androgen receptor (AR), which in turn reduces PSA production. In prostate cancer patient samples, higher miR-379 levels were linked to lower PSA levels and less AR activity. These findings suggest that miR-379 may help prevent prostate cancer from spreading to bones by regulating AR and PSA.

**Abstract:**

**Background:** MicroRNA-379 (miR-379) has been reported to play a tumour-suppressing role in several cancer types. Our previous work demonstrated that miR-379 overexpression attenuates the metastatic spread of prostate cancer (PCa) both in vitro and in vivo. However, the underlying mechanisms remain poorly understood. **Methods**: To elucidate the mechanisms by which miR-379 affects metastases, we performed a cytokine array to identify secreted proteins modulated by miR-379 dysregulation in a bone microenvironment model. We then assessed the levels of the key candidate, and performed functional studies, including reporter assays, of the transcriptional regulation. **Results:** Prostate-specific antigen (PSA)—the clinically widely used blood biomarker for PCa—emerged as the most significantly affected secreted protein. We observed that PSA secretion increased following miR-379 inhibition and decreased with miR-379 overexpression, with parallel changes in intracellular PSA levels. However, our data suggests that miR-379 does not directly regulate PSA expression. Instead, miR-379 appears to downregulate androgen receptor (AR) expression by targeting its 3′-untranslated region (3′-UTR), thereby indirectly reducing PSA transcription through diminished AR-mediated promoter activation. Supporting this indirect mechanism, analysis of clinical samples from prostate cancer patients revealed an inverse correlation between expression of miR-379 in prostatic tissue and serum PSA levels. Furthermore, reduced miR-379 expression was associated with increased levels of AR immunostaining in malignant tissues. **Conclusions**: Taken together, these findings suggest that miR-379 negatively regulates PSA secretion indirectly via suppression of AR, and that the interplay between miR-379, AR, and PSA may contribute to the metastatic progression of PCa to bone.

## 1. Introduction

In European men, prostate cancer (PCa) has the highest incidence of all cancers, and it is the third highest cause of cancer-related deaths after lung and colorectal cancer [1]. At present, prostate-specific antigen (PSA) testing is the standard approach for identifying men with PCa in clinical settings [2]. PSA is a serine proteinase, encoded by the *KLK3* gene, and is a member of the human kallikrein family located on chromosome 19q13.3–13.4 [3]. The expression of PSA is mainly regulated by androgens through the androgen receptor (AR) at the transcriptional level. PSA is expressed in both malignant and non-malignant epithelial cells of the prostate, but the secretion into serum is increased in malignant states, which can then be detected to diagnose and monitor PCa [4]. In blood, the majority of PSA is complexed with protease inhibitors. The clinically used DELFIA assay detects both free PSA (fPSA), which is unbound, and total PSA (tPSA), which includes both the unbound PSA and complexed PSA [5,6]. A decreased ratio of fPSA relative to tPSA is associated with higher risk of PCa in men [5].

Metastatic disease is the primary cause of death for men with PCa. The most common site for metastatic spread is the bone, accounting for 90% of metastases at autopsy [7]. There are three distinct phenotypes of bone metastases. They can be either osteoblastic, osteolytic, or mixed, with approximately 70% being osteoblastic [8]. The relationship between the PCa cells and the bone-forming osteoblasts is critical in the metastatic process; this is why we use these cells in our in vivo models mimicking a bone-like setting.

microRNAs play an essential role in post-transcriptional gene regulation through binding messenger RNA (mRNA) and mainly blocking the translation. Since each miRNA can bind to multiple mRNA targets, dysregulation of miRNA expression is frequently implicated in various diseases, most notably cancer. Therapeutic strategies involving miRNA mimics or inhibitors hold promise in impeding or reversing the progression of PCa.

One especially promising miRNA is miR-379, which is part of a highly evolutionary conserved miRNA cluster, miR-379/miRNA-656, located on chromosome 14. This miRNA cluster is the second largest miRNA cluster in the human genome and has been associated with multiple cancers [9]. Our previous work demonstrated lower levels of miR-379 in PCa bone metastases compared to primary tumours and non-cancerous prostatic tissue [10,11]. Notably, low expression of miR-379 is associated with reduced overall survival in PCa patients, and individuals with metastatic disease presented reduced expression of miR-379 in their primary tumour [11]. Consistent with these observations, inhibition of miR-379 was shown to promote metastatic dissemination to bone, whereas restoring miR-379 levels in mouse models reduced the metastatic burden [10,12]. Similar tumour-suppressive properties of miR-379 have also been described in several other cancers, including lung, breast, and gastric cancers and osteosarcoma [13]. In this study, we set out to further elucidate the molecular mechanism behind the effect of miR-379 on PCa metastatic dissemination in PCa.

## 2. Results

Cytokine arrays were performed on medium harvested from 22Rv1 cells transduced with anti-miR-379 compared to scrambled (Scr) control cells, both grown in osteoblast-conditioned media (OBCM). The results showed that PSA exhibited the largest fold change with increases of 3.91 and 2.49, respectively (Figure 1 and Appendix A). DELFIA assays were used to measure the levels of secreted PSA (both total and free) to validate the cytokine array findings. 22Rv1 cells transfected with anti-miR-379 or Scr were utilised, as were 22Rv1 cells transfected with miR-379 or a negative control. Consistent with the cytokine array findings, decreased levels of miR-379 resulted in significantly higher levels of secreted fPSA and tPSA in both normal medium and OBCM (Figure 2). Conversely, when 22Rv1 cells were transfected with miR-379 and grown in normal medium, fPSA and tPSA were significantly lower as miR-379 expression was increased (Figure 2c). In addition, a DELFIA assay performed on VCaP cells transfected with miR-379 or a negative control in normal medium revealed a similar trend in secretion of fPSA and tPSA (Appendix A).

We next investigated the intracellular levels of PSA to determine if miR-379 was exerting its effect on PSA secretion or production. We found that intracellular PSA protein levels (total and free) were significantly higher when miR-379 was inhibited compared to an Scr control (Figure 3a). This was also seen for fPSA in the OBCM, but there was not a significant difference for tPSA in this setting (Figure 3b). The intracellular PSA expression was lower in miR-379-transfected cells for both fPSA and tPSA, but this was only statistically significant for fPSA (Figure 3c). These results suggest that miR-379 mainly affects the production of PSA.

To determine if miR-379 targets PSA directly, we performed a 3′-untranslated region (UTR) luciferase reporter assay. We found no evidence of miR-379 binding to the 3′-UTR of PSA (Appendix A). We also searched available target prediction sites, but there was no predicted miR-379-5p target site in the PSA encoding *KLK3*-transcript using the DIANA microT webserver or TargetScan 8.0 (7 August 2025). When we performed an Ago2-IP with miR-379-transfected 22Rv1 cells (Appendix A), PSA was not identified as a direct target in this screen, but notably its main regulator AR was enriched in this assay (*p* = 0.03). As PSA is known to be under the control of androgens through AR binding to its promoter region, we performed a luciferase reporter assay with the PSA promoter and found that increased levels of miR-379 led to significantly less binding to the PSA promoter region (Figure 4). This opens the possibility that the regulation of PSA is indirect through AR. To validate this, we performed a Western blot to look at protein expression of AR in miR-379-overexpressing 22Rv1 cells and found that AR expression was decreased in the miR-379-transfected cells (Figure 5).

To assess if miR-379 binds directly to the 3′UTR of AR, we performed a luciferase reporter assay using PC3 cells lacking endogenous AR, minimising background binding. Reporter constructs containing fragment 2–8 of the AR 3′UTR were co-transfected with either miR-379 or scramble mimics. A significant reduction in luciferase signal was observed for construct 2, 6, 7, and 8, indicating that miR-379 can directly regulate AR expression by binding to these parts of the 3′UTR (Figure 6b). AR 3′UTR constructs 2 and 6 were further evaluated in independent biological repeats in PC3 cells and construct 6 was found to be consistently downregulated (Figure 6c). Further, blocking endogenous levels of miR-379 in PNT2 (a normal immortalised prostate cell line with the highest miR-379 levels [12]) gave an increased luciferase signal in construct 6 of the AR 3′UTR, indicating that this interaction can occur in prostate cells (Figure 6d).

To evaluate the clinical relevance of our in vitro findings, we determined the levels of miR-379, AR, and PSA in a cohort of 70 PCa patients. The results from the patient cohort show a significant negative association between PSA levels in the blood and miR-379 expression in prostatic tissue (Figure 7a). Furthermore, malignant prostate tissues with low miR-379 expression were significantly associated with a higher AR score (Figure 7b).

## 3. Discussion

The overarching aim of this study was to enhance understanding of PCa metastasis and to address the unmet clinical need for novel therapeutic strategies. miRNAs are increasingly recognised as regulators of transient cellular processes driving metastatic dissemination. This transient miRNA regulation leads to intratumoural heterogeneity, enabling phenotypic adaptation as cancer cells encounter environmental changes during extravasation, migration, and colonisation at distant sites.

We have previously identified miR-379 as a suppressor of PCa dissemination to bone in vivo and in vitro and shown that PCa patients with bone metastases exhibit lower miR-379 levels, suggesting that miR-379 downregulation facilitates metastatic spread of PCa to bone [10,11,12]. Functional assays confirmed that miR-379 upregulation can inhibit metastatic spread of PCa to the bone and reduce cell growth, migration, colony formation, and adhesion to bone [12]. To elucidate the underlying molecular mechanisms, we performed a cytokine array in which PCa cells were grown in OBCM to mimic the bone microenvironment. PSA emerged as the most strongly upregulated cytokine upon miR-379 downregulation in 22Rv1 cells. This finding was validated by performing DELFIA assays that confirmed increased levels of f- and tPSA when miR-379 was downregulated, and lower levels when miR-379 was upregulated, in both OBCM and standard medium. The PSA levels are somewhat higher in the OBCM, in line with previous reports on osteoblast-induced PSA expression [14], possibly due to osteoblast secretion of IL-6 that can activate AR and stimulate PSA expression [15]. but our results show no drastic difference when comparing the results in normal media to OBCM, suggesting that this mechanism is independent of the factors only present in OBCM. However, intracellular levels of tPSA were not affected by anti-miR-379 in OBCM, it is conceivable that the stability or turnover of PSA bound to ACT is affected by factors in OBCM. However, intracellular levels of tPSA were not affected by anti-miR-379 in OBCM, it is conceivable that the stability or turnover of PSA bound to e.g., alpha-1-antichymotrypsin (ACT) is affected by factors in OBCM. We acknowledge the limitations mimicking the bone environment using OBCM models, as it is only one cell type, whereas the bone comprises a complex microenvironment mainly made up of osteoblasts, osteoclasts, and osteocytes.

Intracellular fPSA measurements revealed the same trend, suggesting that miR-379 affects the expression of PSA through transcription or stability rather than secretion. Prior studies conducted by our group and others showed that miRNAs can directly regulate PSA [16,17,18]; however, neither in silico prediction tools nor the Ago2-IP assay in 22Rv1 cells could identify PSA as a direct target of miR-379. In contrast, AR was enriched in the Ago2-IP following miR-379 overexpression, leading us to hypothesise that the effect of miR-379 on PSA is mediated by AR. Androgens acting through AR or the constitutively active isoform of AR is well established as the main regulator of PSA expression [19,20]. This was further supported by luciferase reporter assays and Western blots, as well as patient data showing that low levels of miR-379 were associated with higher AR score in TURP specimens. These finding suggests a possible feedback loop between miR-379 and AR, as supported by a recent work by Salehi et al. where androgen upregulation stimulated exosomal release of miR-379, reducing intracellular miR-379 levels and enhancing cellular proliferation [21]. We propose a model were decreased levels of miR-379 increase AR activity, which further suppresses miR-379 levels, ultimately increasing PSA levels. Interfering with the androgen signalling pathway is the most effective way to treat PCa, and increasing the miR-379 levels might represent a complementary therapeutic strategy.

Consistent with this, we observed a negative correlation between miR-379 expression in prostatic tissue and serum PSA levels in our patient cohort. Elevated PSA levels could contribute to the osteoblastic phenotype of bone metastases by cleaving parathyroid-hormone-related protein that enhances bone resorption [22], increasing Runx2, involved in osteoblast differentiation [23], and activating TGF-β, a stimulator of new bone formation [24,25]. This could lend support to the “vicious cycle” of PCa bone metastases, where cancer cells lead osteoblasts and resorbing osteoclasts to secrete factors that in turn facilitate the expansion and establishment of metastases.

Although our data demonstrate that PSA can be deregulated by miR-379 in vitro and in patients, the specific functional role requires further clarifications. Prior studies indicate that increased levels of PSA secretion in xenografts enhance cell migration and osteoblastic proliferation in the femur [26]. These previous data together with our data suggest that the miR-379-AR-PSA axis may contribute to metastatic progression and targeting this axis could represent a promising therapeutic strategy to limit PCa metastatic potential. 

## 4. Materials and Methods

### 4.1. Cell Culture and Transfecction

22RV1, PC3, PNT2, and VCaP prostate cancer cells were purchased from ATCC (American Type Culture Collection, Manassas, VA, USA) and cultured as per the supplier’s guidelines. The cell line was validated and tested regularly for mycoplasma (Eurofins scientific, Ebersberg, Germany). Cells were transfected with Miridian Dharmacon hsa-mimic-379-5p (# IH 300687-06-005) or Miridian Dharmacon NEG ctrl 1 (#CN-001000-01-05) at a concentration of 5 nM using Oligofectamine (Invitrogen, Carlsbad, CA, USA) according to the manufacturer’s instructions.

### 4.2. Acquiring OBCM Conditioned Media

Human primary mesenchymal bone marrow stem cells (provided by Professor Stefan Scheding at Lund University) were cultured using StemMACS expansion medium (Miltenyi, Bergisch-Gladbach, Germany) and after establishment differentiated into osteoblasts using low-glucose DMEM with 10% FBS, 10 mM β-glycerophosphate, 0.05 mM l-ascorbic acid, and 0.1 μM dexa-methasone (Sigma-Aldrich, Steinheim, Germany). After three weeks, differentiated osteoblasts were validated with 10 mg/mL Alizarin Red (Sigma-Aldrich) staining.

For experiments using OBCM, medium was collected after 48 h to 72 h, and cell debris was cleared by centrifugation at 900 g; a 1:1 ratio of OBCM and normal growth medium was used for the cell lines and normal media.

### 4.3. Cytokine Arrays

22Rv1 anti-miR-379 cells and 22Rv1 Scr cells were grown in 1:1 OBCM for 72 h in 6-well plates. The medium was then replaced with fresh growth media and the cells incubated for a further 18 h, before the supernatant was collected. The Proteome Profiler Human XL Cytokine Array Kit (BioTek, Winooski, VT, USA) was used, following the manufacturer’s instructions, except for leaving the initial incubation overnight at 4 °C and imaging exposure for 30 s. A cut-off was set at 20 QL/pixel^2^, and cytokines below this threshold were excluded.

### 4.4. Luciferase Assays

PSA Luciferase assays were performed with the 3′UTR sequence of the *PSA* gene cloned into the pMIR-REPORT luciferase vector (Thermo Scientific, Waltham, MA, USA) or pGL3 Enhancer vector (Promega, Madison, WI, USA) containing the PSA promoter as described in Larne et al. [18] (Appendix A). The pRL Renilla firefly construct (provided by Professor Lars Rönnstrand, Lund University) was co-transfected to normalise for variation in transfection efficiency and cell number. Co-transfection of the constructs and anti-miR-379 or negative control mimics was performed. The cells were harvested 24 h after transfection, and firefly luciferase and Renilla luciferase signals were measured with a Wallac 1420 Victor2 reader (Perkin Elmer) using the Dual-Luciferase^®^ Reporter Assay System (Promega, WI, USA).

### 4.5. AR 3′UTR Construct and Luciferase Assay

Due to the length of the AR 3′UTR (almost 7 kb), it has previously been cloned in eight separate pieces into the pMIR-REPORT Luciferase vector (Ambion, Austin, TX, USA) [27,28]. For the luciferase assays, PC3 or PNT2 cells were seeded 24 h before transfection with 800 ng of AR 3′-UTR reported plasmids, 5 ng of pNL1-1 TK control luciferase vector, and either 80 nM of mimic or 80 nM of hsa-mir-379-5p inhibitor /Negativ ctrl A with 2 ul Dharmafect Duo (Horizon). Firefly luciferase and Nano luciferase activity was assayed 48 h after transfection with a Nano-Glo Luciferase Reporter assay system # N1610 (Promega, WI, USA), and chemiluminescence was measured on the plate reader (BioTek).

### 4.6. Western Blots

Cell lysates of the miR-379-transfected cells were collected 48 h post transfection using M-PER lysis buffer (Thermo Scientific), and protein concentrations were determined using a Coomassie Bradford assay (Thermo Scientific, Rockford, IL, USA). A total of 20 µg protein per sample was mixed with 4× Laemmli buffer (BioRad, Hercules, CA, USA) and incubated at 95 °C for 5 min. Samples were then loaded into a 4–20% Mini-PROTEAN^®^ TGX™ gel (BioRad) alongside a molecular weight marker and run until bands had reached the bottom of the gel. Proteins were transferred onto membranes using the Trans-Blot Turbo system (BioRad). The membrane was blocked at 4 °C with 5% milk in PBST, and, after adding the respective antibody, incubated overnight. The primary antibodies used were AR (Santa Cruz, #sc-816, Dallas, TX, USA) diluted 1 in 500 and β-actin (Sigma-Aldrich, #A5441) diluted 1 in 5000. Detection was performed using the Amersham imaging machine Amersham Imager 600 (GE Healthcare), and ImageJ (version 1.54p) was used for densitometry. Uncropped Western blots can be found in Appendix A.

### 4.7. Patient Cohort

RNA was previously extracted from prostate specimens, collected between 1990 and 1999 in Malmö, Sweden, after a transurethral resection of the prostate (TURP) to relieve urinary retention, from 70 men; 47 with PCa and 23 without evidence of PCa [18,29] (Appendix A). The cohort originally consisted of 75 men, but 5 were excluded due to low RNA concentrations. AR immunohistochemistry was performed on slides adjacent to the ones used for the RNA extraction, and PSA was measured in serum from these patients at the time of diagnosis. The study was approved by the Regional Ethical Review Board in Lund, all patients gave written informed consent, and we have adhered to the Helsinki Declaration.

### 4.8. DELFIA Assay

Cells lysates were collected using 10X RIPA Buffer #ab156034 (Abcam, Cambridge, UK). Protein concentrations were determined using the Pierce™ BCA Protein Assay Kit (Thermo Scientific, MA, USA). These were then sent to Mona Hassan Al-Battat at Hans Lilja’s laboratory for DELFIA measurements. Free PSA (fPSA) and total PSA (tPSA) were measured using DELFIA Prostatus™ PSAF/T (Perkin-Elmer Life Sciences, Turku, Finland) [30]. The results were normalised to total protein concentration in the cell lysate as measured by BCA protein assay, according to the manufacturer’s protocol.

### 4.9. RNA Immunoprecipitation of AGO2

Cells were harvested 48 h after transfection with either miR-379 mimics or control, using lysis buffer (20 mM Tris–HCl pH 7.5, 150 mM NaCl, 0.3% Nonidet P-40, 2 mM EDTA, 1 mM NaF). The lysate was shaken for 30 min on ice before undergoing centrifugation at 10,000× *g* for 20 min at 4 °C. The lysate was pre-cleared with protein G sepharose beads 4 Fast Flow (GE Healthcare, Waukesha, WI, USA) for one hour on rotation at 4 °C to reduce background signal. Samples were first incubated with antibody against human AGO2 (Sigma-Aldrich, Saint Louise, MS, USA) on rotation overnight at 4 °C and then incubated with protein G sepharose beads on rotation for 72 h at 4 °C. The protein/RNA complexes were digested with 40 μg Proteinase K (Qiagen, Hilden, Germany) for 30 min at 50 °C. The samples underwent RNA extraction for downstream sequencing.

### 4.10. RT-qPCR to Measure miR-379 Levels

For this test, 25 ng RNA was reverse-transcribed using the qScript Flex Kit (#95049-100, Quantabio, Beverly, MA, USA) in 8 μL, 2 μL 5× reaction mix, 1 μL GSP enhancer, 0.05 μM two-tailed RT primer, and 0.5 μL reverse transcriptase. RT was performed at 25 °C for 1 h, then 85 °C for 5 min, before samples were held at 4 °C. The qPCR was performed with PowerUp SYBR Green Master Mix (#A25742, Thermo Scientific) on the QuantStudio 7 Flex qPCR machine (Applied Biosystems, Pleasanton, CA, USA), as described by Voss et al. [11]. Specific primers for unedited miR-379 were used for RT and qPCR as previously described [11]. The qPCR programme consisted of 30 sec of initial denaturation at 95 °C, and 45 cycles of 5 sec at 95 °C and 20 sec at 60 °C. To exclude the amplification of unspecific products, a melt curve analysis was performed. TaqMan microRNA assays U47 (#001223), RNU48 (#001006), and RNU66 (#001002) (Applied Biosystems, Pleasanton, CA, USA) were used for housekeeping controls according to the manufacturer’s instructions with 1 ng RNA input. The miR-379 levels were normalised to the geometric mean of the housekeeping controls using the ΔCt method.

### 4.11. Statistical Analysis

Statistical analysis was performed using GraphPad Prism 9 (GraphPad software, La Jolla, CA, USA), and statistical significance was determined using a two-tailed unpaired Student’s *t*-test unless stated otherwise. *p* < 0.05 was considered statistically significant.

## 5. Conclusions

In conclusion, our findings identify miR-379 as a modulator of the AR-PSA axis in PCa. Increasing knowledge about the interplay between miR-379 and AR/PSA will increase our understanding of the molecular mechanisms of PCa cancer metastasis and may ultimately facilitate the development of improved biomarker-based detection and monitoring of metastatic spread and novel therapeutic strategies.

## Figures and Tables

**Figure 1 cancers-17-03245-f001:**
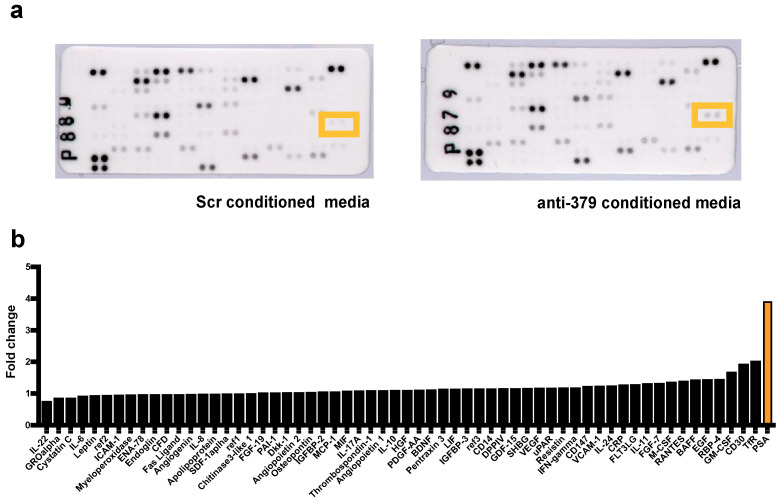
Cytokine array: 22Rv1 cells transduced with anti-miR-379 compared to an Scr control grown in OBCM. The raw blots (**a**) are shown with spots corresponding to PSA highlighted in orange boxes. The waterfall plot (**b**) shows the positive fold change increase in cytokines from 22Rv1 cells grown in OBCM. PSA is highlighted in the orange bar.

**Figure 2 cancers-17-03245-f002:**
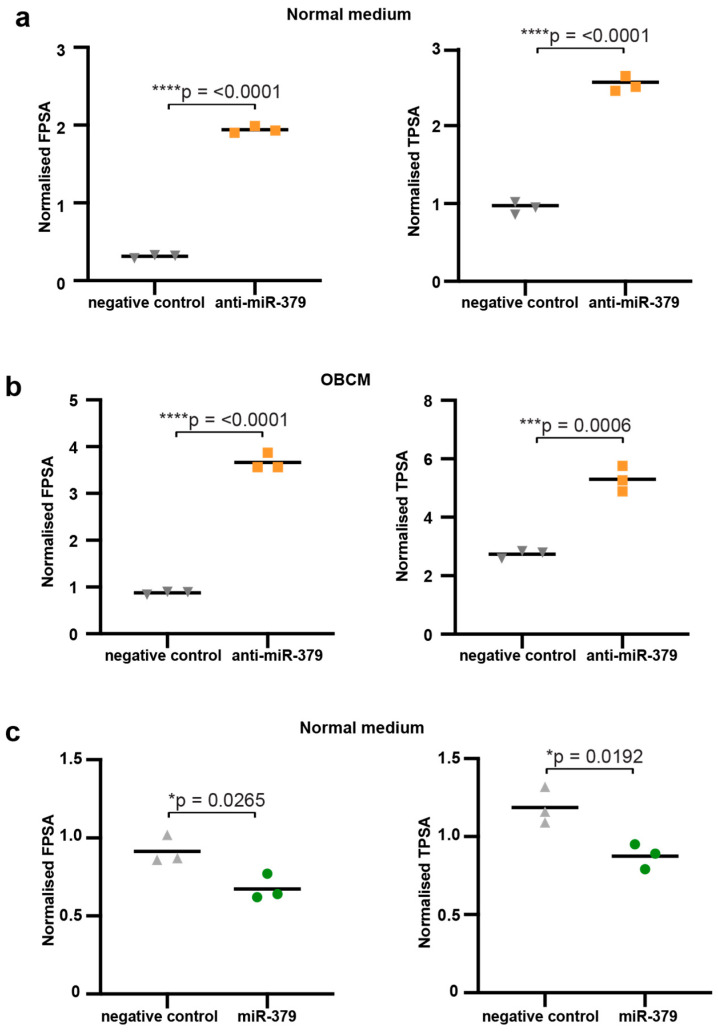
DELFIA secreted PSA. Both total and free PSA were measured in supernatants from 22Rv1 cells. (**a**) The 22Rv1 cells transfected with scrambled control (negative control) are shown to the left in grey, and cells transfected with anti-miR-379 are shown to the right in orange in normal medium and (**b**) in OBCM. (**c**) The 22Rv1 cells transduced with scramble control (negative control) in grey and miR-379 (green circles) in normal media are shown. Unpaired two-tailed Student’s *t*-tests were performed to compare the treatment groups to one another; * *p* < 0.05; *** *p* < 0.001; **** *p* < 0.0001. Only statistically significant *p* values are shown in the figure. Experiments were performed three times, and representative data is shown.

**Figure 3 cancers-17-03245-f003:**
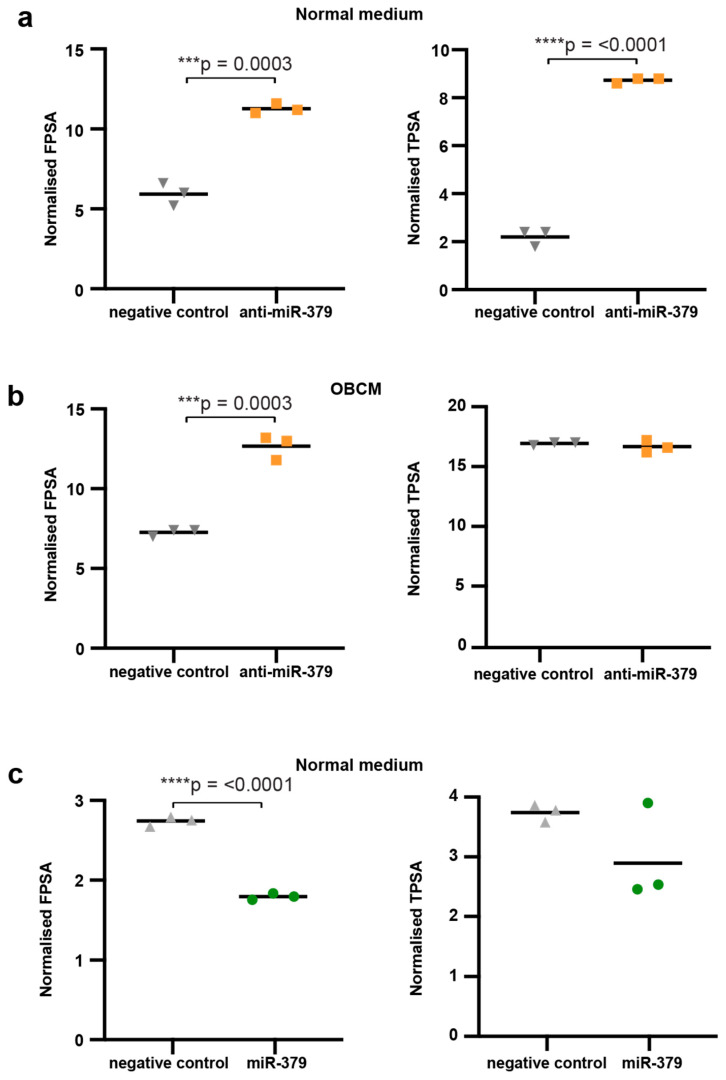
DELFIA intracellular PSA. Both total and free PSA were measured in 22Rv1 cells. (**a**) The 22Rv1 cells transfected with scrambled control (negative control) are shown to the left in grey, and cells transfected with anti-miR-379 are shown to the right in orange in normal medium and (**b**) in OBCM. (**c**) The 22Rv1 cells transduced with scramble control (negative control) in grey and miR-379 (green circles) in normal media are shown. Unpaired two-tailed Student’s *t*-tests were performed to compare the treatment groups to one another; *** *p* < 0.001; **** *p* < 0.0001. Only statistically significant *p* values are shown in the figure. Experiments were performed three times, and representative data is shown.

**Figure 4 cancers-17-03245-f004:**
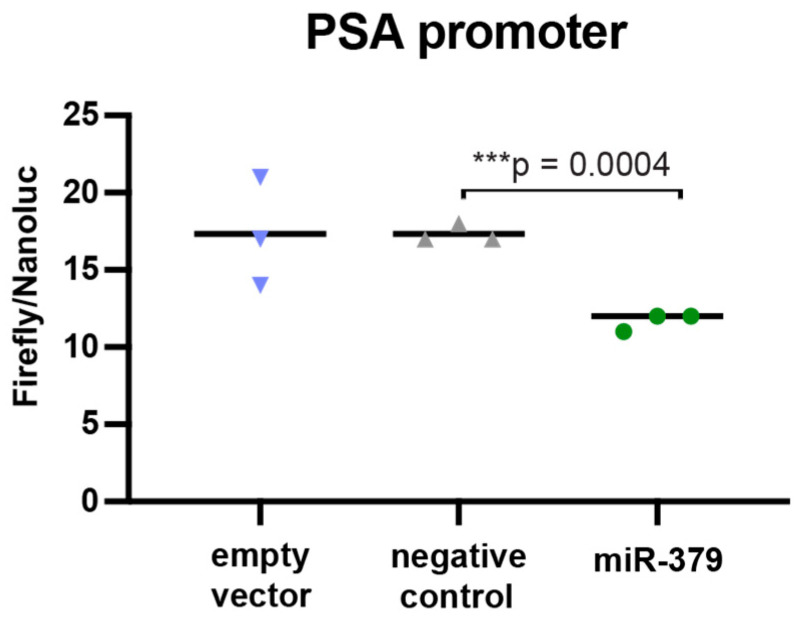
PSA promoter. Luciferase activity was measured in 22Rv1 cells. The left bar denotes the empty vector, negative control cells are in the middle, and the right bar denotes miR-379 cells. Unpaired two-tailed Student’s *t*-tests were performed to compare the treatment groups to one another; *** *p* < 0.001. Only statistically significant *p*-values are shown in the figure. Experiments were performed three times, and representative data is shown.

**Figure 5 cancers-17-03245-f005:**
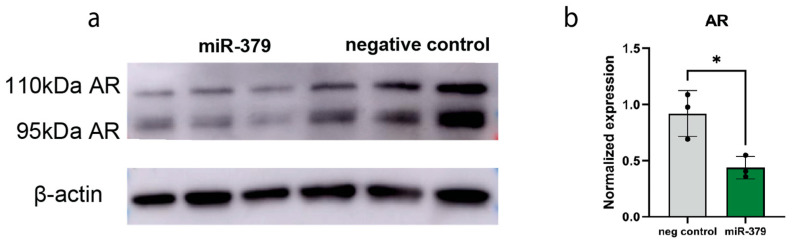
Protein expression of AR; 22Rv1 cells were transfected with miR-379 mimics for 48 h before protein isolation and (**a**) Western blotting. Total expression of AR (110 kDa isoform and 95 kDa isoform) was normalised to the expression of β-actin using densitometry (**b**). Mean and triplicates are shown. Unpaired Student’s *t*-tests were performed to compare the treatment groups to one another, * *p* < 0.05.

**Figure 6 cancers-17-03245-f006:**
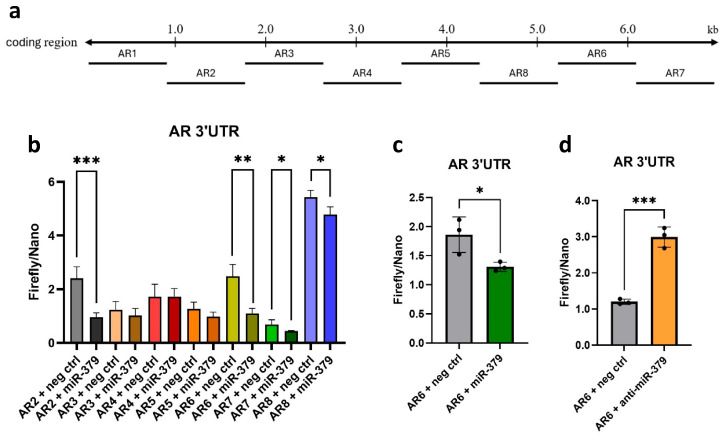
Luciferase reporter assay detection of miR-379 and AR 3’UTR interaction. (**a**) Schematic diagram of the AR 3′UTR in RefSeq (6680 b, ENST00000374690). Short black lines (#1–8) denote the fragments cloned into pMIR-REPORT Luciferase vector. (**b**) PC3 cells were co-transfected with the respective reporter construct and miR-379 or negative control, and the luciferase activity was measured after 48 h incubation with the Nano-Glo Luciferase Reporter assay system. The results were normalised to an empty vector treated the same way. Data shown as average of quadruplicates. (**c**) PC3 cells were co-transfected with AR 3′UTR construct 6 and miR-379 or negative control. The luciferase activity was measured after 48 h incubation with the Nano-Glo Luciferase Reporter assay system. The results were normalised to an empty vector treated the same way. The data shown are averages of three separate experiments, each conducted in triplicate. (**d**) PNT2 cells were co-transfected with AR 3′UTR construct 6 and anti-miR-379 or control. The luciferase activity was measured after 48 h incubation with the Nano-Glo Luciferase Reporter assay system. The data shown are averages of three separate experiments, each conducted in triplicate. Unpaired two-tailed Student’s *t*-tests were performed; * *p* < 0.05; ** *p* < 0.01; *** *p* < 0.001.

**Figure 7 cancers-17-03245-f007:**
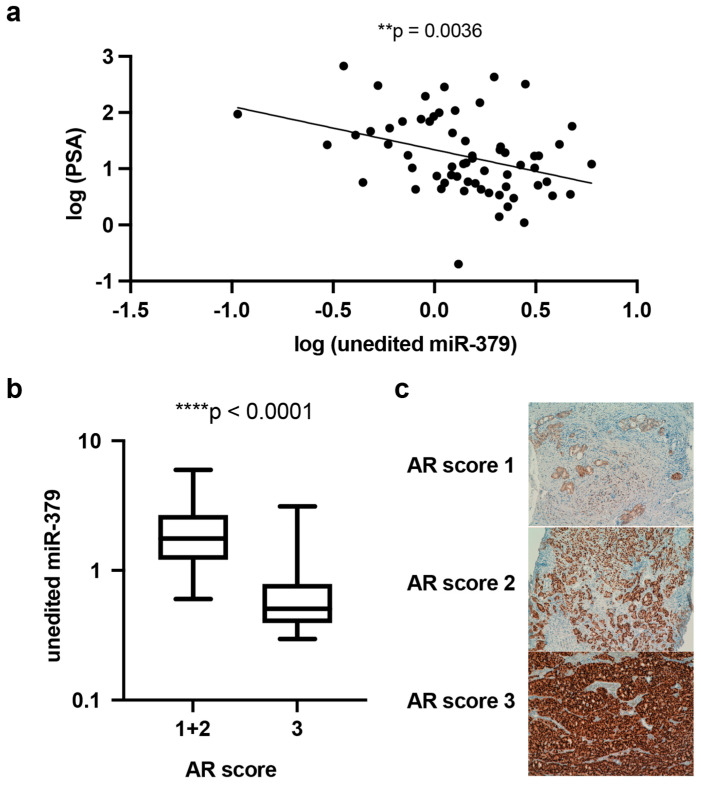
Correlation of miR-379 levels in prostatic tissue and serum PSA levels/AR score in a PCa cohort of 70 men. (**a**) The serum levels of PSA in a cohort of men who had undergone TURP plotted against miR-379 levels in the tumour tissue. The levels of miR-379 measured using qRT-PCR were normalised against the geometrical mean of RNU47, RNU48, and RNU66. Data was log-transformed to enable meaningful linear regression. (**b**) Correlation between miR-379 levels and AR intensity score 1 + 2 versus 3 in malignant epithelial cells in PCa patients. (**c**) Malignant prostatic epithelia were immunohistochemically stained for AR, and the intensity and quantity were scored from 1 to 3 (representative staining shown). RNA was extracted from adjacent tissue slices, and the levels of miR-379 were measured using qRT-PCR normalised against the geometrical mean of U47, RNU48, and RNU 66. A Mann–Whitney U test was used to compare patient groups; ** *p* < 0.005; **** *p* < 0.0001. Only statistically significant *p* values are shown in the figure.

## Data Availability

The raw data supporting the conclusions of this article will be made available by the authors on request.

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
