# Peer review of "MicroRNA-379 Modulates Prostate-Specific Antigen Expression Through Targeting the Androgen Receptor in Prostate Cancer"

_cancers, 2025, doi:10.3390/cancers17193245_

Round 1
Reviewer 1 Report
Comments and Suggestions for Authors
In this study the authors continued their previous work on miR-379, showing that this miRNA has tumor-suppressive properties in prostate cancer preclinical models. The molecular mechanism was, however, not clear.
The authors show that miRNA-379 modulates (suppresses) PSA expression in 22RV1 prostate cancer cells primarily through indirect regulation of the PSA via the androgen receptor. Inhibition of miR-379 resulted in a consistent increase in both total and free PSA. Furthermore, miR-379 overexpressing 22RV1 cells displayed reduced PSA promoter activity. The luciferase reporter assays in the AR-negative PC-3 cells and the enrichment of AR in Ago2-IP experiments strengthen the hypothesis of direct miR-379 binding to the AR 3’UTR, particularly in AR 3’UTR construct 6, which was consistently downregulated across the biological replicates. In patient samples, an inverse correlation between PSA in the serum and miR379 expression levels in the prostate was observed.
Major concerns:
- The majority of the results are shown in only one prostate cancer cell line, namely 22RV1. While the findings in these cells are interesting, relying on a single cell line limits the generalizability of the findings severely. The observed effects of miR-379 could be just cell line-specific. Validation in additional prostate cancer model(s), would strengthen the evidence, for example in the LNCaP or VCaP cell line or preferably in more near-patient PCa organoid models.
- While the AR protein levels showed a decreasing trend with miR-379 mimics in the 22RV1 cells, this was not statistically significant, which weakens the argument for a robust regulatory effect. Moreover, in CRPC PCa patients usually many AR splice variants emerge, that can be used as biomarkers for poor prognosis and resistance to androgen deprivation therapy since these variants are not inhibited by the traditional androgen deprivation therapeutics. What is the effect of the miR-379 on these variants (especially the clinically relevant ARv7?)
Overall, the data suggest a nuanced role for miR-379 in PSA regulation, likely mediated through AR, but further validation in additional cell line(s) as well as for the AR variants is needed to confirm the mechanistic pathway and its clinical relevance.
Minor revisions:
- Although the interactions between the prostate cancer cells and the bone forming osteoblast are critical in the homing part of the metastatic process also multiple other cells in the bone stroma are pivotal to this process. This should be addressed in the introduction.
- In the introduction, the authors mention the three distinct types of bone metastases (osteoblastic, lytic or mixed). Does miR-379 affects this three types differently? Is there any mechanistic effect of miR-379 on why some PCa bone metastasis are lytic and others osteoblastic?
- The DELFIA assay in figure 1 displayed an increase in PSA levels in 22RV1 cells grown in OBCM medium. However, as shown in the supplementary data, the DELFIA assay displayed only a trend in VCaP cells in secretion of free PSA and total PSA when cultured in normal medium. Is this also the case in the OBCM medium?
- It is not clear when the PSA of the patient cohort was measured: was this (as written in the materials and methods) at time of diagnosis or at time of surgery (supplemental table 2).
- Did you observe differential levels of miR-379 in the patients with metastatic prostate cancer vs the patients without metastases? (validation of your previous findings)
- Could you add some information to the supplementary table: Gleason score of the PCa patients, and please elaborate on the type of metastasis in the 25 patients with metastasis? LN or distant?
Reviewer 2 Report
Comments and Suggestions for Authors
This is a high-quality piece of scientific work with a clear and compelling outcome. The study is well-designed, and the conclusions are largely well-supported by the data. It addresses an important mechanism in prostate cancer (PCa) metastasis. The combination of in vitro models (including the bone-mimicking OBCM), luciferase assays, and validation in a clinical cohort is a significant strength. The story is logical and easy to follow: miR-379 down → AR up → PSA up. The link to the bone metastatic niche is a crucial contextual element.
The manuscript is well written. The flow is professional and appropriate for a high-impact scientific journal. I have some minor suggestions for stylistic improvement.
However, I have some critical suggestions for data improvement.
Major Scientific Points
Weakest Link: Western Blot for AR (Figure 5). This is the most significant weakness. The text states there is only "a trend for decreased AR expression," and the result is not statistically significant. This is a critical piece of evidence for the proposed mechanism (miR-379 → AR ↓ → PSA ↓), and its lack of clear evidence undermines the conclusion. My suggestion is to repeat the Western Blot to ensure the experiment is powered sufficiently (have more than 2 biological replicates). Consider using a different, more validated antibody or an alternative method. qRT-PCR for AR mRNA, to strengthen this data, will be useful. If the result remains non-significant, the language in the discussion must be toned down to reflect that this part of the mechanism is suggestive but not definitively proven by the data. The strong luciferase and clinical IHC data help, but the direct protein-level evidence in the authors' cell model is currently lacking, in my opinion.
In addition, the gold standard for proving a linear pathway (A → B → C) is a rescue experiment. The paper shows miR-379 affects AR (via luciferase) and that AR affects PSA (a known fact). To solidify the claim that miR-379 affects PSA through AR, you should attempt to rescue the PSA suppression by co-expressing AR in the miR-379-overexpressing cells. It will be greatly beneficial for the paper to perform a rescue experiment. Transfect cells with: 1) Control, 2) miR-379 mimic, 3) miR-379 mimic + AR expression vector (lacking the 3'UTR so it's immune to miR-379). If PSA levels in group 3 are restored to control levels, it provides very powerful evidence for the authors' proposed mechanism.
About the functional Role of PSA. The discussion on whether PSA is a driver or just a bystander marker of metastasis is interesting but speculative. The statement "decreasing PSA levels may represent a promising therapeutic strategy" is overreaching, as data do not test this directly. I suggest toning it down. I suggest framing it as a hypothesis generated by your findings. E.g., "Our data, combined with the work of Cumming et al., suggest that the miR-379-AR-PSA axis may contribute to metastatic progression, and targeting this axis to reduce PSA secretion could represent a promising therapeutic strategy." This focuses on the axis studied, not just PSA alone.
Minor Points and Clarifications
Figure 2/3 Legends: The legends are good but could be slightly clearer. Instead of "the left bar denotes Scr...", be more direct: "Cells transfected with scrambled control (Scr) are shown on the left, cells transfected with anti-miR-379 are shown on the right."
Introduction Flow: The transition from miRNAs in general to miR-379 specifically is excellent. However, the paragraph on bone metastases ends abruptly. A single sentence linking the bone microenvironment to your experimental model (OBCM) would improve the flow.
Result on Intracellular PSA in OBCM (Fig. 3b): Note that the increase in tPSA upon miR-379 inhibition is not significant in OBCM, unlike in normal media. This is interesting. Briefly comment on why this might be ("This may suggest factors in the OBCM partially modulate PSA stability or turnover independently of miR-379").
Conclusion:
This is a strong paper with a compelling story. Addressing the Western blot data is the most critical improvement. Adding a rescue experiment would make it exceptionally robust. If fixed, this work has the potential to be a notable publication in the PCa field.I recommend publishing this work independently of the question with Figure 5 (Western blot).
If better data for Figure 5 will be obtained, I would strongly recommend publishing.
Author Response
Please see attachement!

Reviewer 3 Report
Comments and Suggestions for Authors
Cassidy et al., reports on the role of microRNA-379 in regulating PSA expression by modulation of AR levels in prostate cancer. They implicate this indirect regulation of PSA in PCa progression and its bone metastasis. The study has a translational potential and is clinically relevant linking molecular biology experiments with patient data. However, several aspects require clarification.
- Could you explain why the cytokine experiment was performed only in OBCM and not normal medium.
- Please add figure legends for the supplemental data to show what the data represents.
- You mention in your discussion that the data in Fig 2a and 2b shows no drastic difference in PSA levels when comparing the results in normal media to OBCM, however, the normalized levels in anti-miR-379 treated cells are much higher in OBCM, can you explain this discrepancy. It would be worthwhile to compare statistically the levels between the two media conditions.
- Consider performing the results in Fig 2c and 3c in OBCM as well.
- Similar to point 3 above, explain the discrepancy in the results in figure 3b and c for TPSA.
- The data is figure 4 and 5 are still just correlative. Could you confirm that AR is the only receptor that binds to PSA promoter. If not consider performing these experiments in AR deficient cells. Or rather optimize your experimental conditions to validate direct binding; Luciferase activity is as a result of AR binding to the PSA promoter.
- What are the different lane samples in Figure 5. If they are replicates, explain the substantial differences between the replicates especially in the negative controls.
Round 2
Reviewer 2 Report
Comments and Suggestions for Authors
The revision of the manuscript and the authors' responses are quite compelling. I agree that the proposal to "attempt to restore PSA suppression by coexpressing AR in cells overexpressing miR-379" will require significant experimental studies and a publication delay. Therefore, I recommend publishing the revised manuscript in its current version.
Reviewer 3 Report
Comments and Suggestions for Authors
The authors have addressed all my concerns. I do not have any more comments.